

# Rapid growth in Late Cretaceous sea turtles reveals life history strategies similar to extant leatherbacks

Laura E. Wilson

Sternberg Museum of Natural History & Department of Geosciences, Fort Hays State University, HAYS, KS, United States

## ABSTRACT

Modern sea turtle long bone osteohistology has been surprisingly well-studied, as it is used to understand sea turtle growth and the timing of life history events, thus informing conservation decisions. Previous histologic studies reveal two distinct bone growth patterns in extant sea turtle taxa, with *Dermochelys* (leatherbacks) growing faster than the cheloniids (all other living sea turtles). *Dermochelys* also has a unique life history compared to other sea turtles (large size, elevated metabolism, broad biogeographic distribution, *etc.*) that is likely linked to bone growth strategies. Despite the abundance of data on modern sea turtle bone growth, extinct sea turtle osteohistology is virtually unstudied. Here, long bone microstructure of the large, Cretaceous sea turtle *Protostega gigas* is examined to better understand its life history. Humeral and femoral analysis reveals bone microstructure patterns similar to *Dermochelys* with variable but sustained rapid growth through early ontogeny. Similarities between *Progostegea* and *Dermochelys* osteohistology suggest similar life history strategies like elevated metabolic rates with rapid growth to large body size and sexual maturity. Comparison to the more basal protostegid *Desmatochelys* indicates elevated growth rates are not present throughout the entire Protostegidae, but evolved in larger and more derived taxa, possibly in response to Late Cretaceous ecological changes. Given the uncertainties in the phylogenetic placement of the Protostegidae, these results either support convergent evolution towards rapid growth and elevated metabolism in both derived protostegids and dermochelyids, or a close evolutionary relationship between the two taxa. Better understanding the evolution and diversity of sea turtle life history strategies during the Late Cretaceous greenhouse climate can also impact current sea turtle conservation decisions.

## INTRODUCTION

The timing of major life history events in sea turtle species is poorly understood because they spend most of their lives at sea (*Bolten, 2003*). This makes devising effective conservation measures particularly difficult. Because osteohistology can be used to assess age, growth rates, skeletal maturity, and sexual maturity, it plays an important role in sea turtle conservation biology. Consequently, the osteohistology of many modern sea turtle

Corresponding author
Laura E. Wilson, lewilson6@fhsu.edu

populations has been surprisingly well studied (*Zug, Wynn & Ruckdeschel, 1986*; *Snover & Hohn, 2004*; *Snover & Rhodin, 2007*; *Avens & Goshe, 2007*; *Braun-Mcneill et al., 2008*; *Goshe et al., 2009*; *Snover et al., 2011*; *Petitet et al., 2015*). Despite this wealth of knowledge regarding bone growth in modern taxa, the long bone osteohistology of fossil sea turtles is virtually unknown. The purpose of this study is to examine the osteohistology of long bones of *Protostega gigas*, a large Late Cretaceous protostegid sea turtle, to better understand its growth dynamics. Framing analyses within the context of known bone microstructure, biology, and ecology in extant sea turtles help elucidate the timing of *Protostega* life history events. Additional comparisons to other extinct protostegid and non-protostegid taxa shed light on the evolution and phylogenetic distribution of sea turtle growth strategies in the Late Cretaceous with possible implications for conservation efforts.

*Protostega gigas* is the second largest known sea turtle taxon (behind its sister taxon *Archelon ischyros*), reaching a length of 3.4 m with a flipper span of 4.7 m (based on DMNH 1999 specimen). For comparison, the leatherback sea turtle (*Dermochelys coriacea*, the largest extant species) grows to lengths around 2 m. Like *D. coriacea*, *P. gigas* had a reduced carapace and plastron. Specimens are found in Santonian to Campanian-aged marine rocks of the Western Interior Seaway and Atlantic coast, with the northern-most definitive specimen from the Pembina Member of the Pierre Shale in Manitoba, Canada (*Nicholls, Tokaryk & Hills, 1990*). Although the phylogenetic position of the Protostegideae in relation to other turtle groups is not clearly resolved (see discussion below), the genera included in and monophyly of the Protostegidae are fairly consistent (*Hirayama, 1994*, *1998*; *Hooks, 1998*; *Kear & Lee, 2006*; *Cadena & Parham, 2015*; *Evers & Benson, 2018*). Within these phylogenetic frameworks, *Protostega* is considered one of the most derived protostegids and sister taxon to *Archelon*, who seemed to replace *Protostega* in late Campanian seas. Historically, several *Protostega* species have been named, including *P. gigas* (*Cope, 1871*), *P. potens* (*Hay, 1908*), *P. dixie* (*Zangler, 1953*), and *P. eaglefordensis* (*Zangler, 1953*). However, *Hooks (1998)* suggested removing *P. eaglefordensis* from the genus and synonymized all remaining *Protostega* species into *P. gigas*, making *Protostega* monospecific. *Hooks's (1998)* taxonomy is followed here.

Because osteohistologic patterns record the history of bone growth for an organism, and bone growth reflects phylogenic, ontogenic, biomechanic, and environmental factors, osteohistology studies can be used to infer life history strategies of extinct organisms (*Cooper et al., 2008*; *Padian & Lamm, 2013*; *Marín-Moratalla, Jordana & Köhler, 2013*). Histologic features like vascular canal density, vascular canal orientation, osteocyte lacunae shape and density, and collagen fiber orientation are used to infer relative growth rates. Cyclical growth marks (CGMs; annuli and lines of arrested growth) are used to calculate absolute growth rates and the age at time of death (see *Padian & Lamm, 2013* for overview). Changes in growth rates through the life of an organism are used to infer life history traits like metabolism, age at sexual maturity, and age at somatic growth (*e.g.*, *Padian & Lamm, 2013*), making histology important for understanding vertebrate growth.

The Cheloniidae (which includes all extant sea turtle species except the leatherback) shows similar osteohistologic patterns. All sampled taxa have low global compactness, indicating overall spongiose bone (*Nakajima, Hirayama & Endo, 2014*). Loggerheads

(*Caretta caretta*) (*Zug, Wynn & Ruckdeschel, 1986*; *Snover & Hohn, 2004*; *Casale et al., 2011*; *Guarino et al., 2020*), Kemp's ridleys (*Lepidochelys kempii*) (*Goshe et al., 2009*; *Snover et al., 2011*), olive ridleys (*Lepidochelys olivacea*) (*Petitet et al., 2015*), and green sea turtles (*Chelonia mydas*) (*Snover et al., 2011*) have a spongiose medullary area that grades into a more compact cortical bone towards the periosteal surface. Cortical bone is characterized by small, longitudinal vascular canals arranged in concentric rows. The size and density of vascular canals typically decrease towards the periosteal surface with the thickness of the dense periosteal cortical bone increasing with ontogeny (*Snover & Hohn, 2004*; *Snover et al., 2011*; *Guarino et al., 2020*). Secondary remodeling is present in older individuals (*Zug, Wynn & Ruckdeschel, 1986*; *Snover & Hohn, 2004*). While not all previous studies have noted the collagen fiber orientation associated with cheloniid bones (since many studies are focused on skeletochronology and samples are often decalcified), some authors note loggerheads have parallel-fibered bone (*Zug, Wynn & Ruckdeschel, 1986*; *Houssaye, 2013*).

Though not as well studied as some of the other extant taxa, leatherbacks have a distinctly different bone microstructure. Global compactness profiles reveal an even greater degree of spongiose bone and vascularity compared to cheloniids (*Kriloff et al., 2008*; *Nakajima, Hirayama & Endo, 2014*; *Houssaye, Sander & Klein, 2016*). Large vascular canals are longitudinally oriented and arranged in concentric rows, but sampled individuals lack the denser cortical bone on the periosteal margin (*Rhodin, 1985*; *de Ricqlès, Castanet & Francillon-Vieillot, 2004*). Similar bone microstructures are observed in the humerus, femur, and tibia, despite differences in function between the fore- and hindlimbs. Because researchers have used decalcified or micro-CT scanned bones, collagen fiber organization has not been noted. The difference in leatherback bone growth is particularly intriguing considering the unique biology and ecology of leatherbacks with rapid early ontogenetic growth, elevated body temperatures, gigantothermy, deep diving capabilities, and fully pelagic lifestyles (*Lutcavage & Lutz, 1986*; *Paladino, O'Connor & Spotila, 1990*; *Spotila, O'Connor & Paladino, 1997*; *Bolten, 2003*). Some studies estimate the age at sexual maturity for leatherbacks as early as 6 years (*Rhodin, 1985*) or averaging around 13 to 15 years for females (*Parham & Zug, 1997*; *Dutton et al., 2005*; *Jones et al., 2011*). While others calculate leatherback sexual maturity closer to 20 years old (*Avens et al., 2009*), most estimates are younger than for cheloniid turtles, which can take up to 50 years to reach sexual maturity at a smaller maximum body size (see *Jones et al., 2011* and references therein).

Within the Protostegidae, only the bone histology of *Archelon ischyros* (the sister taxon to *Protostega*) has been noted. *Rhodin (1985*: 763) briefly described the microstructure of a phalange as "nearly identical to the pattern in the leatherback" with no clear transition between medullary and cortical regions and no compact cortical bone. Although *Archelon* shell has been histologically sampled (*e.g.*, *Scheyer & Sanchez-Villagra, 2007*), no other extinct sea turtles have had long bones histologically studied and changes in bone microstructure through ontogeny are not well understood for extant or extinct sea turtles. The lack of rigorous study leaves many questions regarding the osteohistologic patterns

and their relationship to the life history strategies of protostegids, specifically, and extinct sea turtle taxa in general.

## MATERIALS AND METHODS

### Institutional abbreviations

CM—Carnegie Museum of Natural History, Pittsburgh, Pennsylvania, USA; DMNH—Denver Museum of Nature and Sciences, Denver, Colorado, USA; FHSM—Fort Hays State University's Sternberg Museum of Natural History, Hays, Kansas, USA; KUVP—University of Kansas Museum of Natural History, Vertebrate Paleontology Collection, Lawrence, Kansas, USA.

### Materials

Fossil specimens histologically sampled for this study are listed in Table 1. Different sized *Protostega* specimens were selected with the goal of capturing multiple ontogenetic stages. Because studies on modern sea turtles show growth and life history differences between geographically separate populations (*e.g.*, *Seminoff et al., 2002*; *Bjorndal et al., 2003*, *2013*; *Chaloupka, Limpus & Miller, 2004*; *Balazs & Chaloupka, 2004*; *Peckham et al., 2011*; *Ramirez et al., 2020*; *Avens et al., 2020*), only *Protostega* specimens collected from the Smoky Hill Member of the Niobrara Formation of Kansas were used in this study.

Additional taxa were also sectioned for comparison. *Desmatochelys lowi* provides an example of long bone microstructure in a more basal protostegid and *Toxochelys* is generally considered an outgroup to all other sea turtle clades (*Kear & Lee, 2006*; *Cadena & Parham, 2015*; *Raselli, 2018*; *Scavezzoni & Fischer, 2018*; *Gentry et al., 2018*; *Evers & Benson, 2018*; *Evers, Barrett & Benson, 2019*). Six previously sectioned *Dermochelys* individuals were loaned from Dr. Anders Rhodin (Chelonian Research Foundation) for analysis. See *Rhodin (1985)* for slide preparation methods. Descriptions of extant cheloniids included in this study are based on previous publications.

### Methods

Humeri and femora were thin sectioned to assess ontogenetic stage and growth rates (Fig. 1). Humeri were sectioned just distal to the deltopectoral crest and femora were sectioned at the mid-diaphysis. *Zug, Wynn & Ruckdeschel (1986)* sectioned multiple elements from loggerhead cranial, axial, and appendicular skeletons to access suitability for osteohistology analysis. Of the bones sectioned (carapace, dentary, cervical vertebrae, phalanx, ulna, and humerus), the authors found the humerus most suitable due to the preservation of growth marks in periosteal bone, but also state that the femur is likely suitable, as well. The authors also show that the highest density of cortical bone with the least amount of resorption and remodeling in the humerus was located just distal to the deltopectoral crest. Subsequently, sectioning humeri just distal to the deltopectoral crest is common practice in sea turtle osteohistology studies (*e.g.*, *Zug, Wynn & Ruckdeschel, 1986*; *Snover & Hohn, 2004*; *Goshe et al., 2020*) and was followed in this study (Fig. 1). The FHSM VP-17979 femur was sectioned at the mid-diaphysis, as is typical in most tetrapod long bone ontogentic studies (*e.g.*, *Padian & Lamm, 2013*).

**Table 1 Fossils used for osteohistological analysis in this study.**

| Taxon | Museum number | Element | Length (cm) | CGMs |
|---|---|---|---|---|
| *Protostega* | | | | |
| | FHSM VP-17979 | Humerus | 18.0 | 4 |
| | FHSM VP-17979 | Femur | 14.2 | 4 |
| | CM 1393 | Humerus | 17.7 | 2 |
| | CM 1421 | Humerus | 33.8 | 8* |
| | KUVP 1208 | Humerus | 35.0 | 8* |
| *Desmatochelys* | | | | |
| | FHSM VP-17470 | Humerus | | 15+ |
| *Toxochelys* | | | | |
| | FHSM VP-700 | Humerus | 14.5 | 5+ |

**Note:**

* Total CGMs estimated by retrocalculation.

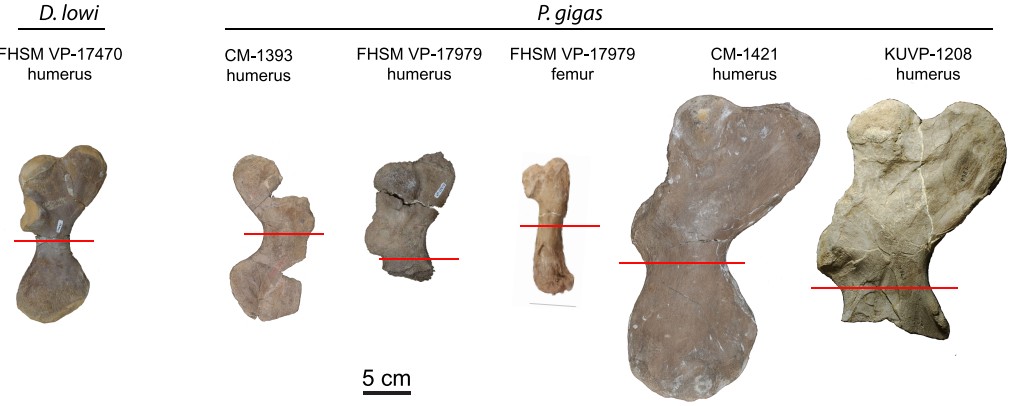

**Figure 1 Protostegid sea turtle long bones histologically sampled.** One long bone from the basal protostegid *Desmatochelys lowi* and four long bones from the derived protostegid *Protostega gigas* were thin sectioned for analysis and comparison. Red lines indicate where samples were taken. See Table 1 for absolute bone sizes and text for institutional codes and sampling protocols.

    All specimens were photographed, molded, cast, and 3D scanned prior to sectioning. Osteohistology methods followed *Lamm (2013)*. A tile saw was used for most sectioning, except for FHSM VP-700, FHSM VP-17979 femur, and CM-1393, which were sectioned with an Isomet Low Speed saw. Most bones were embedded in Silmar 41 with an MEKP catalyst; the FHSM VP-17979 femur was embedded in Buehler EpoThin epoxy resin and hardener. All sections were mounted to glass slides with either Devcon 2-Ton Epoxy or J.B. Weld ClearWeld. Sections were ground to optical clarity on a Buehler Ecomet lap wheel. Thin sections were only made and observed in transverse section.

    As bone expands, primary bone is absorbed and remodeled into secondary bone, permanently obscuring the early ontogenetic record, including the presence of any CGMs. Taphonomic alteration like crushing and bacterial invasion can also obscure the growth record, especially in the spongiose medullary region of sea turtle long bones. Consequently, qualitative retrocalculation was used to estimate any potential missing CGMs from the

inner regions of the sampled long bones to allow for a more accurate age estimate of individuals at the time of death. To estimate lost CGMs due to secondary remodeling and taphonomic alteration, smaller and larger humeri were appropriately scaled and the smaller specimen was transposed on the larger in Adobe Photoshop. The number of CGMs identified in the smaller humerus but missing for the larger was added to the number in the larger specimen to estimate the age of the larger individual at the time of death.

Slides were analyzed using an Olympus BX53M transmitting microscope with a polarizer, and photographs were taken with an Olympus SC180 camera. Images were edited using Olympus Stream Essentials and Adobe software. 3D surface scans and photogrammetry models of sectioned specimens are reposited on Morphosource (Project ID: 000418396); high resolution images of thin sections are reposited in MorphoBank (Project 4289).

## RESULTS AND DISCUSSION

*Protostega* long bone osteohistology has never been described, so detailed descriptions of sampled specimens are provided in the Supplemental Material. In general, similar histologic patterns are observed in all *Protostega* bones analyzed in this study. Well-vascularized spongiose bone with abundant, round osteocyte lacunae, mixed woven, parallel-fibered, and lamellar bone, and widely spaced CGMs provide evidence of sustained, rapid growth during all sampled ontogenetic stages (Fig. 2, Figs. S1 and S2). Parallel-fibered and lamellar bone is predominantly concentrated around primary osteons, with woven bone making up the matrix between osteons. Given the bone fiber matrix and primary osteons, *Protostega* bone is characterized as fibrolamellar (*sensu de Ricqles et al., 1991*). An external fundamental system (EFS) is characterized by closely spaced CGMs, low vascularity, flattened osteocytes, and/or lamellar bone along the periosteal surface. When present, it indicates somatic maturity. No EFSs are observed in any sampled *Protostega* bones, meaning that even the older *Protostega* individuals (CM-1421 and KUVP-1208) had not reached skeletal maturity in their ninth year (Figs. 2E and 2F, Fig. S2). Because sexual maturity and decreased growth rates are closely correlated in sea turtles (*Wood & Wood, 1980*; *Price et al., 2004*; *Casale et al., 2009*; *Avens & Snover, 2013*; *Bjorndal et al., 2014*; *Omeyer, Godley & Broderick, 2017*; *Turner Tomaszewicz et al., 2022*), no *Protostega* individuals had likely reached sexual maturity at the time of death either. Despite this, the humerus grew to over 35 cm in length by age eight and doubled in length between ages four and eight (Table 1). One of the largest recorded *Protostega* humeral lengths is 42 cm from a Mooreville Chalk specimen (*Renger, 1935*; *Danilov et al., 2022*). If this specimen represents a skeletally (and thus sexually) mature individual, then KUVP 1208 (the largest specimen in this sample set) is 85% of maximum humeral length. With sustained growth rates, it is possible that *Protostega* reached skeletal and sexual maturity within 10 years.

Previous studies reveal two osteohistologic patterns in extant sea turtle populations. All sampled cheloniid taxa have low global compactness, indicating overall spongiose bone (*Nakajima, Hirayama & Endo, 2014*). Leatherbacks display extremely spongiose bone throughout the cortex with no clear separation between the medullary cavity and cortical
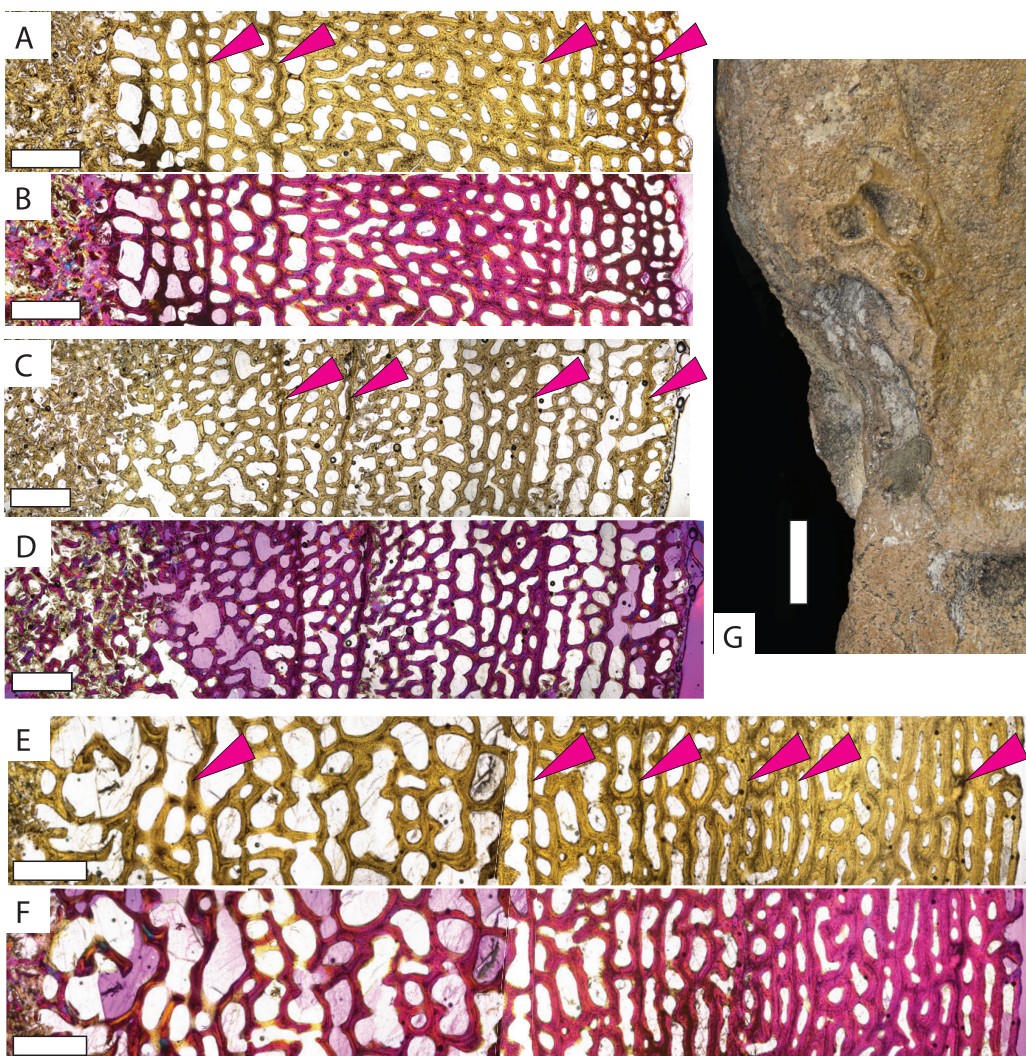

**Figure 2 Micro- and macrostructures observed in *Protostega gigas* long bones.** Small *P. gigas* humerus FHSM VP-17979 in (A) plane light and (B) polarized light with a lambda filter. Small *P. gigas* femur FHSM VP-17979 in (C) plane light and (D) polarized light with a lambda filter. Note the irregularly spaced cyclical growth marks (pink arrows) in both the humerus and femur. Large *P. gigas* humerus KUVP 1208 in (E) plane light and (F) polarized light with lambda filter with pink arrows highlighting CGMs. (G) Epiphysial surface of large *P. gigas* humerus CM 1421 showing rugosities associated with fast-growing vascularized cartilage. Periosteal surface to the right in (A–F). Scale bars on (A–F) is 1 mm; scale bar on (G) is 1 cm.     

bone (*Rhodin, 1985*; *de Ricqlès, Castanet & Francillon-Vieillot, 2004*; *Snover & Rhodin, 2007*; *Kriloff et al., 2008*; *Nakajima, Hirayama & Endo, 2014*; *Houssaye, Sander & Klein, 2016*) (Fig. 3C, Fig. S4). Cheloniids also have low global compactness (*Nakajima, Hirayama & Endo, 2014*), but the outer cortex is denser with lower vascularity even in earlier ontogeny (*e.g.*, *Zug, Wynn & Ruckdeschel, 1986*; *Goshe et al., 2009*; *Casale et al., 2011*; *Snover et al., 2011*; *Petitet et al., 2015*; *Şirin & Başkale, 2021*). When compared to modern sea turtle long bones, *Protostega* long bone microstructure is more similar to leatherbacks, with no distinguishable medullary cavity and highly vascularized bone

extending to the periosteal surface. Even in the oldest individuals sampled, spongiose bone is evidence through the entire cross section, with the denser cortical bone observed in cheloniids lacking.

The similarities between *Protostega* and leatherback osteohistology invites comparison in life history strategies. Leatherbacks differ from other sea turtles in their large body size, completely pelagic ecology, migration into cold arctic waters, deep diving, and continuous swimming (*Paladino, O'Connor & Spotila, 1990*; *Spotila, O'Connor & Paladino, 1997*). One of the most notable leatherback life history characteristics is their elevated resting metabolic rates and ability to hold a body temperature above the surrounding water temperature (*Paladino, O'Connor & Spotila, 1990*; *Spotila, O'Connor & Paladino, 1997*). While they are not considered endothermic, the term 'gigantothermy' was first used to describe the adult leatherback's elevated metabolism (*Paladino, O'Connor & Spotila, 1990*); although, it should be noted that smaller-bodied juvenile leatherback also have elevated resting metabolic rates and unique behaviors like constant swimming (*Lutcavage & Lutz, 1986*). Some of these life history strategies are reflected in bone microstructure. For example, rapid leatherback growth and elevated metabolic rates are denoted in the highly vascularized bone with widely-spaced CGMs. A pattern also observed in *Protostega* (Fig. 2, Figs. S1 and S2).

Studies on leatherback appendicular bones also reveal unique surficial features. The epiphyseal articular surface of leatherback bones has a rough, dimpled subchondral surface that is evidence of highly vascularized epiphyseal cartilage (*Rhodin, Ogden & Conlogue, 1981*; *Rhodin, 1985*; *Snover & Rhodin, 2007*). This unique chodro-osseous characteristic likely reflects high vascularization of cartilage related to rapid growth to large body size (*Rhodin, Ogden & Conlogue, 1981*; *Rhodin, 1985*; *Snover & Rhodin, 2007*). Because this feature is missing in the large, extinct, freshwater *Stupendemys* (*Rhodin, 1985*), it cannot be attributed to large size alone. While this chondro-osseous growth pattern is not seen in other living sea turtles, it has been identified in the derived protostegid *Archelon* (*Rhodin, 1985*). Although vascularized epiphyseal cartilage was originally noted as absent in *Protostega* (*Snover & Rhodin, 2007*), the present study provides evidence that these epiphyseal rugosities are, in fact, present in large *Protostega* humeri (Fig. 2G). These rugosities are absent in the more basal *Desmatochelys*, an observation in agreement with previous reports. *Snover & Rhodin (2007)* suggest that the presence of this character possibly supports a close phylogenetic relationship among protostegids and leatherbacks. While osteohistology does not always directly correlate to phylogeny (*e.g.*, *de Ricqlès et al., 2008*) and this relationship has yet to be widely supported (see phylogenetic discussion below), similarities in osteohistological and chondro-osseous growth patterns between *Protostega* and leatherbacks support similar growth and life history patterns—specifically, rapid growth to large body size with elevated metabolic rates.

The growth pattern observed in *Protostega* bones is in strong contrast to the more basal protostegid *Desmatochelys* (Fig. 3A, Figs. S4A–S4C). *Desmatochelys* humerus microstructure is more similar to cheloniids and the extinct *Toxochelys* (Fig. 3B, Figs. S3D–S3F), having a discernable cortical bone with reduced vascularization and more

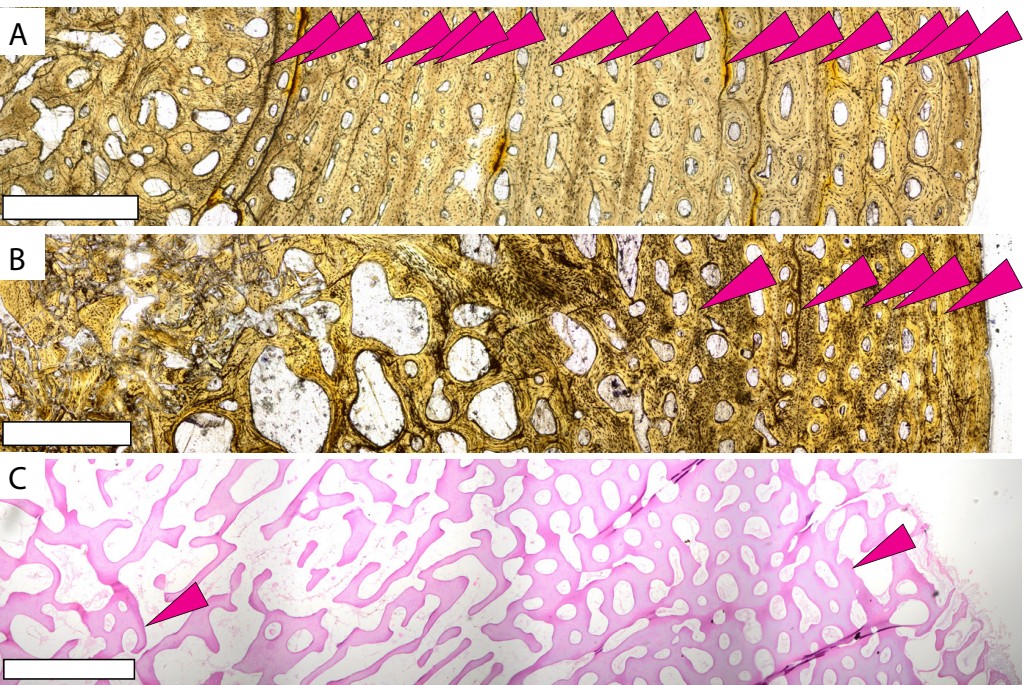

**Figure 3 Microstructures observed in non-*Protostega* sea turtle long bones for histologic comparison.** Humeri of basal protostegid *Desmatochelys lowi* FHSM VP-17470 (A) and non-protostegid sea turtle *Toxochelys latiremis* FHSM VP-700 (B) in plane light. Humerus of modern *Dermochelys corticea* CRF (Chelonian Research Foundation) 4911 (C) in plane light. CRF 4911 is a female of unknown age with curved carapace length of 135 cm. Periosteal surface is to the right in all figures. Scale bar in all figures is 1 mm.

closely spaced CGMs. These histologic patterns indicate prolonged growth with a later ontogenetic attainment of sexual maturity at smaller body size. Consequently, at some point between *Desmatochelys* and *Protostega*, protostegids evolved rapid growth rates to larger size. While it is always difficult to ascertain the biotic and abiotic pressures leading to evolutionary novelties, large body size is a successful defense against predation (*e.g.*, *Reimchen, 1991*; *Chase, 1999*; *Isbell, 2005*). To this point, adult leatherbacks have few non-human predators, owing to their large body size (*James, Myers & Ottensmeyer, 2005*). Though both *Protostega* and *Desmatochelys* are in the Protostegidae and share an evolutionary history, *Desmatochelys* is a more basal taxon that evolved in the late Early Cretaceous (Barremian) tens of millions of years prior to *Protostega*, with *D. lowi* occurrences dating to the Cenomanian and Turonian (*López-Conde et al., 2019*). This puts the origin of the *Desmatochelys* genus prior to the evolution of mosasaurs and polycotylid plesiosaurs (*e.g.*, *Polcyn et al., 2014*; *Fischer et al., 2018*; *Madzia & Cau, 2020*). On the other hand, the appearance of *Protostega* in the Santonian parallels the evolution of large pelagic tylosaurid mosasaurs like *Tylosaurus prorigor*. Organismal and theoretical studies suggest that rapid growth evolves in response to predation pressure (*e.g.*, *Arendt, 1997*; *Arendt & Reznick, 2005*; *Cooper et al., 2008*; *Woodward et al., 2015*), allowing prey taxa to reach a large body size faster and avoid predation at various ontogenetic stages. In the case of Late Cretaceous protostegids, changes in ecosystem structure between the evolution of

*Desmatochelys* and *Protostega* may have resulted in selective pressures favoring the evolution of faster growth rates and larger body size. The evolutionary timing of rapid growth to large body size observed in derived protostegids supports a hypothesis that this growth strategy provided an advantageous evolutionary response to the evolution of large open ocean predators. Increased growth rate would be particularly important for sea turtles where the timing of sexual maturity is correlated with body size (*Omeyer, Godley & Broderick, 2017*), as sexual maturity would also be attained earlier in ontogeny.

Despite overall sustained rapid growth, most individuals in this study show variable bone deposition rates between CMGs. This is most notable in FHSM VP-19797, in which the femur and humerus both preserve similarly uneven bone apposition rates. In this individual, the first and second CGMs are much more closely spaced than the second and third (Figs. 2A–2D). Modern sea turtles are known to have irregular growth rates sometimes depositing closely-spaced CGMs in early ontogeny followed by more widely spaced CGMs later in ontogeny (*e.g.*, *Snover & Hohn, 2004*). Growth plasticity, as illustrated by annual changes in bone apposition rate, would increase survival during periods of harsh environmental conditions (abnormal temperatures, low food availability, *etc.*) (*Köhler et al., 2012*) and would have given *Protostega* an evolutionary advantage over animals with more rigid growth strategies unable to adjust for environmental stress. This phenomenon has been observed in archosauromorphs (*e.g.*, *Cullen et al., 2014*; *Zanno et al., 2019*; *Woodward et al., 2020*), and is now proposed for *Protostega* and perhaps contributing to the overall evolutionary success of sea turtles in general.

Analysis of *Protostega* osteohistology also has interesting phylogenetic implications. While several studies have addressed fossil sea turtle phylogenetics (*e.g.*, *Hooks, 1998*; *Kear & Lee, 2006*; *Joyce, 2007*; *Cadena & Parham, 2015*; *Raselli, 2018*; *Evers & Benson, 2018*; *Evers, Barrett & Benson, 2019*), consensus regarding the phylogenetic placement of various sea turtle taxa, including the Protostegidae, is lacking. Most recent analyses place *Toxochelys* outside and basal to the Chelonoidea, which then includes all extant sea turtles and the Protostegidea (*Kear & Lee, 2006*; *Cadena & Parham, 2015*; *Gentry et al., 2018*; *Raselli, 2018*; *Scavezzoni & Fischer, 2018*; *Evers, Barrett & Benson, 2019*; but see *Gentry, Ebersole & Kiernan, 2019*; *Evers & Benson, 2019*). While leatherbacks are generally regarded as a separate evolutionary lineage from the Cheloniidae, the relationship among leatherbacks, cheloniids, and protostegids is not clear. Most studies resolve *Dermochelys* and the Protostegidae as sister groups (*Hirayama, 1998*; *Kear & Lee, 2006*; *Cadena & Parham, 2015*; *Scavezzoni & Fischer, 2018*) or as a single lineage (*Gentry et al., 2018*), but some align *Dermochelys* more closely with the Cheloniidae (*Raselli, 2018*; *Gentry, Ebersole & Kiernan, 2019*; *Evers, Barrett & Benson, 2019*) leaving protostegids a more distant lineage. A few studies resolve the Protostegidae further removed from other sea turtles as more basal eucryptodires (*Joyce, 2007*; *Anquetin, 2012*). Most studies focused specifically on protostegid phylogentics support Protostegidae and Dermochelyidae as sister groups (*Hooks, 1998*; *Hirayama, 1998*; *Kear & Lee, 2006*; *Cadena & Parham, 2015*; *Scavezzoni & Fischer, 2018*).

If protostegids and dermochelids are sister taxa (*Hirayama, 1998*; *Kear & Lee, 2006*; *Cadena & Parham, 2015*; *Scavezzoni & Fischer, 2018*), then highly spongiose bone and

rapid growth until sexual maturity either evolved convergently in sea turtles or were shared by a common ancestor and *Desmatochelys* secondarily lost this character. Because these bone microstructure patterns are also paired with the presence of vascularized cartilage and rugosities marking the epiphesial surface of the proximal humerus, and this morphological pattern is seen in leatherbacks, *Archelon*, and *Protostega* but not more basal protostegids (*Rhodin, 1985*; *Snover & Rhodin, 2007*; this study), it is likely that other basal protostegids lack the bone growth patterns of *Protostega* and leatherbacks. Consequently, bone microstructure and macrostructure better support the hypothesis that rapid growth strategies are convergent between derived protostegids and *Dermochelys*. Likewise, if *Protostega* and *Dermochelys* are more distantly related (*Joyce, 2007*; *Anquetin, 2012*; *Raselli, 2018*; *Evers & Benson, 2018*; *Gentry, Ebersole & Kiernan, 2019*; *Evers, Barrett & Benson, 2019*), then the osteohistologic patterns seen in these two taxa must be convergent, as other taxa do not share their bone growth pattern (*e.g.*, *Toxochelys*, cheloniids). Alternatively, at least one study has hypothesized that the Dermochelyidae is within the Protostegidae (*Snover & Rhodin, 2007*; *Gentry et al., 2018*). In this case, the similar histologic patterns could be explained by a single evolutionary innovation inherited from a common derived protostegid ancestor. These hypotheses can be tested with more sampling from fossil Dermochelyidae and basal Protostegidae taxa, in addition to refining phylogenetic analyses.

## CONCLUSIONS

Extant sea turtles display two bone microstructure patterns that appear to relate to life history strategies (*Bolten, 2003*; *Snover & Rhodin, 2007*) with leatherbacks having evolved sustained rapid growth to large body size and an elevated metabolism compared to cheloniids. When compared to extant sea turtle osteohistology, the bone microstructure of the Late Cretaceous protostegid *Protostega gigas* more closely resembles leatherbacks than sampled members of the Cheloniidae. Consequently, histological evidence supports a hypothesis that *Protostega* likely shared life history traits with leatherbacks, such as elevated early ontogenetic growth and possibly elevated resting metabolic rates. This is corroborated by the first evidence of vascularize cartilage on the epiphysial surface of the *Protostega* humerus, a character also associated with rapid growth (*Rhodin, Ogden & Conlogue, 1981*; *Rhodin, 1985*; *Snover & Rhodin, 2007*). Results from this study also illustrate that *Protostega* could reach 85% of the body size of the largest known individual within nine years of hatching.

Because the more basal protostegid *Desmatochelys* lacks the same rapid growth patterns as *Protostega*, this evolutionary character likely evolved along the protostegid lineage and is not plesiomorphic to the clade. This has phylogenetic implications regarding the single or multiple origin of rapid growth patterns in large-bodied sea turtles, depending on the phylogenetic placement of *Dermochelys* with respect to the Protostegidae. Regardless, rapid early ontogenetic growth to large body size provides an evolutionary advantage for these sea turtles sharing a pelagic habitat with large predators like sharks, ichthyodectids, plesiosaurs, and mosasaurs.

Although more research is needed, studying the evolution, growth strategies, and biodiversity of extinct sea turtles has implications for extant sea turtle conservation, particularly in light of warming ocean temperatures. Numerous sea turtle taxa thrived in Late Cretaceous oceans under greenhouse conditions, providing a model of the future through the lens of deep time experiments. Exploring the diversity of sea turtle growth strategies, possible environmental stressors leading to evolutionary innovations, and survivability of taxa (with an understanding of their life history strategies) across space and time has the potential to inform sea turtle conservation efforts.

## ACKNOWLEDGEMENTS

Thank you to Matt Lamanna and Amy Henrici (Carnegie Museum of Natural History) for access to CM 1393 and CM 1421 and to Chris Beard and Megan Sims (KU Biodiversity Institute, Natural History Museum) for access to KUVP 1208 for histologic sampling. Anders Rhodin (Turtle Conservancy, Chelonian Research Foundation) gracious loaned his *Dermochelys* thin sections for analysis and comparison. Aly Baumgartner (Fort Hays State University, Sternberg Museum of Natural History) 3D scanned and molded and casted specimens prior to thin sectioning; Matthew Eads (Fort Hays State University) provided 3D images using photogrammetry for smaller specimens. Ted Vlamis, Hannah Hutchinson, Logan White, and Riley Stanford (Fort Hays State University) assisted with histologic preparation of specimens. Holly Woodward (Oklahoma State University Center for Health Sciences) is thanked for valuable discussions throughout the project. Marine Eugenia Pereyra, Jérémy Anquetin, and Torsten Scheyer provided excellent comments that improved the manuscript. Lastly, a special thank you to the late Curtis Schmidt (Fort Hays State University, Sternberg Museum of Natural History) for always sharing his enthusiasm for science, nature, and sea turtles.

### Funding

This study was supported by Faculty Development and Undergraduate Research Experience funds to Laura E Wilson from Fort Hays State University. The funders had no role in study design, data collection and analysis, decision to publish, or preparation of the manuscript.

### Grant Disclosures

The following grant information was disclosed by the authors:
Faculty Development and Undergraduate Research Experience.
Fort Hays State University.

### Competing Interests

The authors declare that they have no competing interests.

## Author Contributions

- Laura E. Wilson conceived and designed the experiments, performed the experiments, analyzed the data, prepared figures and/or tables, authored or reviewed drafts of the article, and approved the final draft.

## Data Availability

All fossil specimens used in this study are reposited in public museums. Repositories may be contacted for accession and detailed specimen information: FHSM (Fort Hays State University's Sternberg Museum of Natural History), KUVP (University of Kansas Natural History Museum, Vertebrate Paleontology collection). CM (Carnegie Museum).

3D surface scans of sectioned specimens are reposited on Morphosource (Project ID: 000418396):

- FHSM:vp:17979 Protostega gigas, https://doi.org/10.17602/M2/M473631
- FHSM:vp:17470 Desmatochelys lowii, https://doi.org/10.17602/M2/M473597
- FHSM:vp:700 Toxochelys sp., https://doi.org/10.17602/M2/M473529
https://www.morphosource.org/projects/000418396/about?locale=en.

High resolution images of thin sections are available in MorphoBank: Project 4289, DOI 10.7934/P4289.

https://morphobank.org/index.php/Projects/ProjectOverview/project_id/4289

## Supplemental Information

Supplemental information for this article can be found online at http://dx.doi.org/10.7717/peerj.14864#supplemental-information.

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
