# Peer review of "Rapid growth in Late Cretaceous sea turtles reveals life history strategies similar to extant leatherbacks"

_PeerJ, doi:10.7717/peerj.14864_

## Round 0.1 · original submission · Minor Revisions

Thanks a lot for this interesting submission. Your manuscript was reviewed by three experts who all have made interesting suggestions for minor to moderate revisions. I strongly encourage you to follow these suggestions and revise your manuscript accordingly. Please also note that the three reviewers and myself have attached annotated PDFs with some additional comments.

Please, reply to the different points raised by the reviewers in your response.

·

Basic reporting

Professional article structure, figures, tables. Raw data shared: I just suggest adding some labels to the figures

Experimental design

no comment

Validity of the findings

no comment

Additional comments

This manuscript is original and very interesting. The paper provides important data which contributes to the field. Since most of the published studies based on osteohistology sea turtles are focused mainly on skeletochronology, deep analysis of histological features as primary bone tissue type and vascularization are welcome. I have a few questions and comment on the file and I suggest to publish after some changes.

Reviewer 2 ·

Basic reporting

• The author confuses throughout the manuscript (incl. suppl info) bone microstructure that reveals insights into life style (e.g., shallow marine- pelagic) with the growth record/growth pattern that reveals insights into life history traits.
 Bone microstructure refers to bone compactness, i.e. spongious vs compact bone (bone mass increase/decrease etc), or simply put the “cavities” in periosteal (vascular canals and resorption/erosion) and endosteal domain (medullary cavity/,medullary region)
 The growth record refers to the count of growth marks but includes also the organization of tissue (loosely organized or high organized structural fibers) and the vascular density (high, moderate, poor/low, avascular etc)
This must be rephrased/corrected throughout! It is also misleading parts of the discussion.
Do not get me wrong bone microstructure can also reveal important insight but its something different than growth !!!

• The histological description of the tissue and vascularization is not adequate in the main manuscript. Its partially mentioned in the suppl. info but main bone tissue type of specimens belongs also in the main text as well as figures of histo details such as woven bone. Please note tha figures in polarized light are much more informative as those with pol and gypsum filter.

• I further miss a summary of the description of life history traits deduced from histology (and in general known) for Cheloniids (line 46 pp or line 98). you did this roughly for Dermochelys (also some more info in terms of years are welcome if not known state so) but not Cheloniids. For the comparison you want to do its important for the reader to know what are the life history traits (in years and others) what is visible in the growth record (and beyond > what is known by observation etc)

• You state that the similarities in bone microstructure and tissue type and growth pattern can point to phylo relationships. Yes, its sometimes possible but usually histology is highly/mainly influenced by function and not by phylo (also sometime possible) (Ricqlès et al. 2008 and many other/newer studies)
• please see also comments in the pdf

for definitions of terms, the problem of woven bone etc, histo is more influenced by function than phylo…
see
Francillon-Vieillot, H., de Buffrenil, V., Castanet, J., Geraudie, J., Meunier, F.J., Sire, J.Y., Zylberberg, L. & de
Ricqles, A. (1990). Microstructure and mineralization of vertebrate skeletal tissues. Pp. 471–530 in: Carter, J.G. (ed), Skeletal Biomineralization: Patterns, Processes and Evolutionary Trends. Van Nostrand Reinhold, New York.
Buffrénil, V. de, Zylberberg, X., de Ricqlès, A., and Padian, K. (eds.): Vertebrate Skeletal Histology and
Paleohistology. CRC Press

De Ricqles, A.J., Padian, K., Knoll, F. & Horner, J.R. (2008). On the origin of high growth rates
in archosaurs and their ancient relatives:complementary histological studies on
Triassic archosauriforms and the problem of a “phylogenetic signal” in bone histology. Annal.
Pal., 94, 57–76.

Experimental design

see above

Validity of the findings

see above

Additional comments

see above and attached pdf

Annotated reviews are not available for download in order to protect the identity of reviewers who chose to remain anonymous.

·

Basic reporting

The author studies the bone microstructure and growth record of long bones of a selection of extinct sea turtles, with focus on the second-largest species Protostega gigas.
The restriction to long bones as the focus of the present paper is not yet clearly laid out in the text, which should be amended accordingly.
There seems also to be a slight misunderstanding of the usage of the terms "growth patterns" (related to growth record, cyclical growth mark formation, dimensions of growth zones, bone deposition rates, skeletochronology, etc.) and "osteohistology/bone microstructure" (related to bone tissue types, fiber orientation, vascularisation patterns, osteocyte shapes and densities, etc.), which can be easily addressed.
Also, the resolution of the supplementary figures could be higher, but it is not clear to me whether this is linked with a preview-stage version or if the images are generally of lower resolution. In either case I encourage the author to add full scale page-width overview images of the sectioned specimens to allow the reader to better understand the presence of cyclical growth marks in the bones.
Finally, bone histology of shell bones has been reported for many of the extinct taxa used in the present study as well - I leave it up to the author if or how these data should be referenced in the text as well.

Experimental design

The author uses standard osteohistology to analyze the fossilized long bones of extinct sea turtles, in comparison to modern forms. This approach is adequate for the study and research question

Validity of the findings

The interpretations are supported by the data provided and the images are overall of good quality. In some cases (see comments in PDF), however, I would give slightly deviating (ie, higher) cyclical growth mark counts as those reported in Table 1.

Additional comments

The study presents novel and important data on the histology/microstructure and the growth patterns of some extinct marine turtles and as such, the article merits publication. There are a few spelling mistakes that should be taken care of, please refer to the annotated pdf.

---

## Round 0.2 · accepted · Accept

Thanks a lot for this careful revision of your manuscript based on the comments made by the three reviewers. I am pleased to inform you that your work can now proceed toward publication in PeerJ.